# A-DenseUNet: Adaptive Densely Connected UNet for Polyp Segmentation in Colonoscopy Images with Atrous Convolution

**DOI:** 10.3390/s21041441

**Published:** 2021-02-19

**Authors:** Sirojbek Safarov, Taeg Keun Whangbo

**Affiliations:** 1Department of IT Convergence Engineering, Gachon University, Sujeong-Gu, Seongnam-Si, Gyeonggi-Do 461-701, Korea; sirojbeksafarov@gmail.com; 2Department of Computer Science, Gachon University, Sujeong-Gu, Seongnam-Si, Gyeonggi-Do 461-701, Korea

**Keywords:** semantic segmentation, convolutional neural networks, colonoscopy, polyp segmentation, deep learning, attention, dilated convolution

## Abstract

Colon carcinoma is one of the leading causes of cancer-related death in both men and women. Automatic colorectal polyp segmentation and detection in colonoscopy videos help endoscopists to identify colorectal disease more easily, making it a promising method to prevent colon cancer. In this study, we developed a fully automated pixel-wise polyp segmentation model named A-DenseUNet. The proposed architecture adapts different datasets, adjusting for the unknown depth of the network by sharing multiscale encoding information to the different levels of the decoder side. We also used multiple dilated convolutions with various atrous rates to observe a large field of view without increasing the computational cost and prevent loss of spatial information, which would cause dimensionality reduction. We utilized an attention mechanism to remove noise and inappropriate information, leading to the comprehensive re-establishment of contextual features. Our experiments demonstrated that the proposed architecture achieved significant segmentation results on public datasets. A-DenseUNet achieved a 90% Dice coefficient score on the Kvasir-SEG dataset and a 91% Dice coefficient score on the CVC-612 dataset, both of which were higher than the scores of other deep learning models such as UNet++, ResUNet, U-Net, PraNet, and ResUNet++ for segmenting polyps in colonoscopy images.

## 1. Introduction

The third most common form of cancer worldwide for both men and women is colorectal cancer, and its prevalence is increasing every year [1]. The primary cause of colorectal cancer is the growth of glandular tissue in the colonic mucosa. Precise and earlier determination of polyps from virtual colonoscopy screenings is of great significance for the avoidance and timely treatment of colon cancer [2]. However, manual detection depends on proficient endoscopists, and it takes a long time. Recent surveys have shown that more than 25% of polyps in patients undergoing colonoscopy are not detected [3]. The late diagnosis of missed polyps can lead to a low survival rate for colon cancer patients [4]. Computer-aided detection (CAD) systems are used to detect and segment polyps from endoscopic images and video screenings, which allows endoscopists to focus their attention on the polyps displayed on the screen and act as a second viewer. This can decrease the likelihood of overlooked polyps [5].

Designing an accurate CAD system is challenging because of the high cost of labeled medical datasets for training and testing. Polyps have a wide range of colors, sizes, shapes, appearances, or combinations of these features. There are similar inter-classes and various intra-classes for four different polyp classes: Adenoma, hyperplastic, serrated, and mixed. In addition, background objects are very similar; for example, the background mucosa can mix with a polyp or stool [6]. Even though these factors make the polyp segmentation task challenging, we surmise that there is still a great prospect to create such systems for medical use.

In recent years, deep-learning-based techniques have achieved significant success in the computer vision domain [7,8,9], and interest in applying deep learning to endoscopic image segmentation has grown. In particular, encoder-decoder-based methods such as U-Net [10], UNet++ [11], SegNet [12], and fully convolutional networks (FCNs) [13] have been commonly used for semantic segmentation. These networks down-sample the image several times to capture the required feature maps and up-sample once or multiple times to enable effective localization [10,13,14]. Furthermore, skip connection strategies have been successful in saving fine-grained information and improving the efficiency of the network, even on complicated datasets.

Recent research has shown that the attention mechanism has been commonly used to preserve the dependency of features in certain computer vision tasks such as object detection [15], image classification [16,17], and image segmentation [18,19,20,21]. The attention method enables the model to attend more closely to essential features without any external supervision, and it can avoid identical feature maps at various scales to lead to better feature representation. The attention mechanism improves network efficiency over traditional methods with or without multiscale features.

In summary, this study makes the following contributions:We designed a new robust U-Net-based encoder-decoder network structure that uses dense connections as a powerful encoder model and accomplishes an adaptable image segmentation algorithm to integrate deep and superficial features, which can directly combine multiscale features to boost segmentation performance.We utilized an attention mechanism that fuses derived information from various modules and focuses on core information by removing noise and irrelevant regions.Our method uses dense blocks, residual blocks, transition blocks, and atrous convolution block capabilities, and it improves the outcome of the colorectal polyp segmentation compared to other state-of-the-art methods. Our model obtained good results with small datasets.We evaluated our model on the Kvasir-SEG and CVC-612 datasets, and the experimental results show that it achieved the highest intersection over union (IoU) and Dice coefficient.

The remainder of this paper is organized as follows. In Section 2, we review some existing related studies. In Section 3, we present our proposed densely connected deep learning architecture. We present the experimental settings and qualitative and quantitative analysis of the semantic segmentation results in Section 4, and Section 5, respectively. In Section 6, we discuss the experimental results. Finally, we conclude this paper in Section 7.

## 2. Related Work

Over the past two decades, the detection and classification of gastrointestinal (GI) tract diseases and the creation of effective, robust methods to automatically detect polyps in colonoscopy images and videos have been active scientific areas. The performance of machine learning-based polyp detection and segmentation software has come close to that of high-level endoscopists.

Some earlier studies used the texture and color details of polyps to create handcrafted descriptors [22,23,24,25]. For instance, Karkanis et al. [22] utilized a supplemented sliding window scheme and color wavelet texture information as descriptors to designate polyps from colonoscopy images and videos. Subsequently, researchers used spatio-temporal, edge, intensity, and shape features to detect polyps automatically. For example, Hwang et al. [26] used elliptical shape information to detect polyps automatically, whereas Wang et al. [27,28] presented edge cross-section profiles. To improve the detection performance, some methods combine two or more features [1,29]. Tajbakhsh et al. [29] integrated local intensity variation patterns and global geometric constraints to detect polyps. Although these methods achieved significant progress, they still suffer from inferior detection accuracy. The primary cause of the low accuracy level is the limited representation ability of handcrafted features to deal with both the low-level inter-class variety between hard mimics and polyps and the high-level intra-class variety of polyps.

Recent deep convolutional neural networks (CNNs) have shown noticeably better results in many biomedical image analysis domains, including object detection [30,31,32,33], classification [34,35,36], and semantic segmentation [10,21,32,37,38]. Some researchers have attempted to use CNNs to manage the automated polyp detection domain. For instance, Tajbakhsh et al. [39] suggested a CNN architecture for polyp detection that takes low-level handcrafted information as input and utilizes a group of CNNs to learn the shape, color, and temporal features of polyps. However, this model learned temporal and spatial information using various networks that may limit the discrimination capability. Therefore, most features from colonoscopy videos have not been fully explored.

The new generation of CNNs uses transposed convolution layers to generate a probability map in image segmentation tasks. Long et al. [13] proposed the fully convolutional network (FCN) method, which achieved state-of-the-art semantic segmentation results. FCN obtains the segmentation results without post-processing steps by using pixel-to-pixel and end-to-end training. Ronneberger et al. added modifications and extensions to the FCN to develop the U-Net [10] architecture. U-Net integrates high-resolution spatial feature maps with high-level contextual information for medical image segmentation. Inspired by these approaches, several researchers have proposed models to solve segmentation issues in a wide variety of areas [14,21,40,41,42].

The majority of studies published in the sphere of polyp segmentation achieved significant results only on special datasets, and test cases often utilized small validation and training datasets [27,43]. Furthermore, some of the scientific work focuses only on a particular type of polyps, and some of them employ non-public datasets, which makes it difficult to compare and reproduce the results. Consequently, the ML models cannot yet achieve similar or better results than endoscopists. There is an opportunity to enhance the efficiency of CAD systems, making major improvements and producing more effective and reliable architectures for polyp segmentation.

## 3. Proposed Method

The A-DenseUNet architecture is based on UNet++ [41] and densely connected convolutional networks (DenseNets) [44], utilizing the strength of U-Net [10] and DenseNet [44]. The proposed A-DenseUNet architecture takes advantage of dense blocks, atrous convolution, residual blocks, attention blocks, and restrictive skip connections.

### 3.1. Overview

The proposed segmentation architecture utilizes the U-Net [10] concept which includes an encoder block on the left and a decoder block on the right. Figure 1 depicts the entire structure of the proposed method. As shown in Figure 1, we changed the original U-Net’s encoder part with DenseNets which allows us to train a deeper network without vanishing gradient. In addition, we added redesigned skip connections that can allow us to make adaptive networks and transfer missed features to the decoder side. Attention block determines which part of the network require more attention in neural network and it also reduces the computational cost of encoding the information in each polyp image into a vector of each dimension. These and other methods are presented below the sections extensively.

The proposed model takes a training dataset X that consists of N sample images x: X = $x_{1}$,$\;x_{2}$,…,$\;x_{N}$, with corresponding Y = $y_{1}$,$\;y_{2}$,…,$\;y_{N}$. Then, each ground truth pixel i of any given sample y is y ∈ [0, 1]. We feed our network with a 224 × 224 × 3 image and obtain a 224 × 224 × 1 output segmentation mask. In the encoding path, an input image passes through a dense block that includes a combination of atrous convolutional, rectified linear unit (ReLU), and batch normalization layers. The dense block is followed by a transition block that contains a pooling layer that reduces the size of the feature map after each successive dense block. In the decoding path, transposed convolution is used to increase the feature map size back to the original size. After a very deep encoder path, there may be a loss of essential details. To handle such a problem, UNet++ [41] introduces restrictive skip connections that combine the encoding part with the output of up-sampling through channel concatenation.

We applied skip connections to unify various depths of U-Nets into one structure and used an attention mechanism to filter irrelevant information from the features. The depth of the network is s = 5, which means we used a down-sampling approach five times and halved its feature map size each time. After five down-samples, we obtained the final 7 × 7 spatial feature map. We used the attention mechanism to create a relationship between the various model information at different depths. The attention blocks reduce the noise and unnecessary features, and only important information can pass to the next layer. The output of the attention block is up-sampled by transposed convolution and concatenated with the same depth output as the encoder part. After concatenation, the feature map passes the residual block (dilated convolution followed by batch normalization and ReLU activation), which allows features to converge more quickly. Other decoder blocks at levels s = 2 to s = 5 use such blocks. Finally, feature maps from all U-Net depths are agglomerated and then averaged, after that 1 × 1 convolution and sigmoid activation, as shown in Equation (1), are used to obtain the final segmentation map. We trained our network with the binary cross-entropy loss function based on the ground truth for the training images. Equation (2) shows the formula of the loss function, where *L* is the loss for a prediction $y_{i}$ consisting of N pixels at a specific output of the network:(1)$$y=\frac{1}{1+e^{-x}}$$
(2)$$L=-\overset{N}{\underset{i=1}{\sum }}y_{i}\left(\log y_{i}-\left(1-y_{i}\right)\log\left(1-y_{i}\right)\right)$$

### 3.2. Dense Units

Training deeper neural networks can increase a model’s accuracy, but it can also cause degradation problems and interrupt the training process [40,44]. To solve this type of problem, Huang et al. [44] proposed densely connected convolutional networks (DenseNets), which allow all subsequent layers to connect directly, as shown in Figure 2.

Accordingly, the $l^{th}$ layer takes the feature maps of all previous layers, $x_{0}$,…$\;x_{l-1}$, as input:(3)$$x_{l}\;=\;H_{l}([x_{0},\;x_{1},\ldots ,\;x_{l-1}])$$
where [$x_{0}$,$\;x_{1}$,…,$\;x_{l-1}$] refers to the concatenation of the feature maps produced in layers 0, …, l − 1. Dense encoder blocks have several advantages. For example, densely connected layers have fewer output dimensions than other networks, which can help to avoid learning excessive features and reduces the time required.

We used DenseNets [44] as the encoder part of our proposed method, densely connected layers provide maximum gradient flow, and very deep neural networks alleviate the vanishing gradient problem. An original DenseNets [44] implemented 121, 169, 201, 264 layer network with the growth rate of k = 32. We implemented various DenseNet layers and growth rates. Finally, we achieved a better segmentation mask with 164 layer DenseNets and a growth rate of k = 48. Table 1 presents the encoder layers of the proposed architecture. The input layer takes 224 × 224 × 3-sized images; thus, all training data were resized to fit the given size.

In the first layer of the network, using 7 × 7 dilated convolution with 96 filters and a stride of two, we obtained an output feature map of 112 × 112 × 96 after the first convolution layer. Then, we used four dense blocks and three transition blocks to create the remaining encoder layers, s = 2 to s = 5. Each dense block includes batch normalization, dilated convolution, and ReLU non-linearity and is repeated several times, as shown in Table 1, to create a deeper encoder path and obtain more robust feature maps. After each dense block (except dense block 4) we put a transition block, which consists of 1 × 1 convolution and 2 × 2 average pooling with a stride of two. A 1 × 1 convolution is used before the pooling layer to reduce the channels of the feature map. After dense block 4, a robust 7 × 7 feature map is obtained, which is decoded to produce the final output.

### 3.3. Adaptive Network Structure

The original U-Net obtains the final segmentation result from a fixed number of down-samples and a corresponding number of up-samples. In practice, some datasets contain images of various sizes, and there is a significant difference between the amount of information contained in various-sized images. One down-sampling and one up-sampling might be sufficient to obtain satisfactory segmentation results from small, simple data. Large, complicated datasets require multiple up- and down-samplings to obtain semantic feature maps of various regions because it is difficult to obtain global information from a small-scale network. Figure 3 represents the U-Net network with depths of one and two. To overcome different depth problems, Zhou et al. [11] proposed redesigning the skip connections to integrate the advantages of different depths of U-Net into one architecture.

We redesigned the skip connections in our proposed method to connect different depths in the U-Net structure. In addition, we added an efficient feature map transition and aggregated different layer characteristics. As shown in Figure 1, we added horizontal dense connections and connections between each depth. These connections allow us to transfer missed features to the decoder side, and to build an adaptive network. Horizontal densely connected layers are equipped with dilated convolution, batch normalization, and residual blocks. Before transfer features from one level to another, we employ an attention mechanism, which enhances the flow of the spatial feature map to the next level of the decoder side. The final segmentation mask is achieved by averaging the output of each U-Net depth and employing a 1 × 1 convolution and sigmoid classifier.

### 3.4. Attention Units

Over the last few years, the attention mechanism has become very popular in various deep learning research areas, starting with natural language processing (NLP) [45]. Recently, it has been applied to computer vision tasks. The attention model has been utilized as a pixel-wise prediction model in the semantic segmentation domain [46]. It identifies the sections of the network that need more attention. Continuous use of an attention mechanism at each level allows long-range spatial dependency of feature maps. The attention block also decreases each image’s computing cost to a fixed dimensional vector. Therefore, the fundamental value of an attention unit is that it is straightforward and can be applied to every input scale to strengthen the consistency of the features that emphasize the result.

Despite the tremendous success of the skip-connection approach in U-Net for handling spatial information, there are still drawbacks that need to be fixed. After downsampling, features are passed to the next stage, and the same features transferred to the decoder side using a skip-connection approach. Therefore, there is a significant semantic gap between the encoder and decoder side feature maps. Further, the noises and irrelevant features are also easily transferred to the next side. As a result, the long-range object relationships in the image cannot be captured, which may cause severe problems in preserving spatial details, especially object boundary information in the network.

To solve this problem, we employed an attention block in the proposed method for medical image segmentation. To better transfer the spatial information to the next stage in the decoding side, we placed the attention block before the up-sampling layer at each level of the U-Net decoder path. Specifically, the model encodes various semantic feature maps at various stages. The attention mechanism is used to enhance the flow of the spatial feature map to the next level of the decoding side; to generate relevant feature maps, up-sampling information is fused with the corresponding encoder-side information. Thus, attention blocks at various stages allow the proposed network to encode low-level to high-level information at different scales and provide only relevant regions to the next layer. Further, the model generates astonishing results by formulating the attention block, which suppresses undesirable features like an artifact, specular reflection, noises, and, hence, preserving long-range contextual feature dependencies.

### 3.5. Dilated Convolution

The concept of dilated convolution comes from wavelet decomposition [47]. It is also called “atrous convolution” and “algorithm à trous.” Dilated convolution enables the model to arbitrarily expand the filter field of view at every DCNN (Deep Convolutional Neural Network) layer. In order to hold both the calculation and the number of parameters contained, CNNs usually use small convolutional kernels (typically 3 × 3). Dilated convolution with a rate of r adds r-1 zeroes between the consequent filter values, as shown in Figure 4. It thus provides an efficient field of view control mechanism and finds the optimal trade-off between detailed localization (small field of view) and assimilation of context (large field of view).
(4)$$\left(F*k\right)\left(p\right)=\underset{s+t=p}{\sum }F\left(s\right)k\left(t\right)$$
(5)$$\left(F{*}_{l}k\right)\left(p\right)=\underset{s+lt=p}{\sum }F\left(s\right)k\left(t\right)$$

We employed dilated convolution to systematically aggregate multi-scale contextual information without losing resolution. Dilated convolutions allow the network to increase the receptive field without adding computational complexity or increasing the network information capability. Besides, dilated convolution allows us to detect fine-details by processing inputs in higher resolutions, and it also broader view of the input to capture more contextual information. We replace normal convolutions from Equation (4) by atrous convolution in Equation (5) with a dilation rate of 2 for every layer of the network. F is a discrete function and k is a discrete filter size. The discrete convolution operator * can be defined as Equation (4). Dilated convolution ${*}_{l}$ can be defined as Equation (5) *l* is a dilation factor, *s* + *lt* = *p* indicates that some points have been skipped during the convolution.

## 4. Experiments

We evaluated our proposed A-DenseUNet architecture on two public segmentation datasets, Kvasir-SEG [48] and CVC-612 [49].

### 4.1. Datasets

Kvasir-SEG: We utilized the Kvasir-SEG dataset [48], which has 1000 polyp images and their corresponding ground truth masks annotated by professional gastroenterologists from Vestre Viken Health Trust in Norway, as shown in Figure 5. The images have sizes ranging from 332 × 487 to 1920 × 1072 pixels, but training and testing were performed with an image resolution of 224 × 224 pixels. The images were randomly split into 80% for training, 10% for validation, and 10% for testing.

CVC-612: In addition, we used the CVC-612 [49] dataset, this database built in collaboration with the Hospital Clinic of Barcelona, Spain. CVC-612 has been generated from 23 different video studies from standard colonoscopy interventions with white light. CVC-612 database comprises 612 polyp images of size 576 × 768 pixels from 31 colonoscopy series. The images were split into training, validation, and testing sets in the ratio of 80:10:10. All training, validation, and testing were performed with an image size of 224 × 224 pixels. Figure 6 shows some example images and corresponding masks from the CVC-612 dataset.

### 4.2. Data Augmentation

The effectiveness of deep learning networks depends significantly on the size of the training dataset. It is clear that in the case of polyp segmentation, the training dataset is limited, at least with respect to typical training images employed in the context of deep learning. Furthermore, certain polyp forms are not represented in the dataset, and for other types only a few examples are available. Hence, it is important to extend the training dataset by data augmentation. Data augmentation is conducted to provide additional polyp images for training deep neural networks. Even though this approach cannot produce new polyp forms, it can provide extra data samples based on various image acquisition conditions, such as colon deformations, camera position, and illumination.

All training samples were resized to 224 × 224 pixels in a manner such that the image aspect ratio was retained. This process included random cropping augmentation. All images were augmented using four augmentation techniques: (1) Rotation, with the angle of rotation randomly chosen from the range 0° to 90°; (2) reflection, horizontally and vertically; (3) elastic deformation with a fixed 10 × 10 grid; and (4) color adjustment by random gamma augmentation. After augmentation, the Kvasir-SEG training dataset contained a total of 8000 images.

### 4.3. Evaluation Metrics

To evaluate the polyp segmentation, we used the following well-known segmentation evaluation metrics: recall, precision, intersection over union (IoU), and Dice coefficient. We calculated these metrics using well-known parameters, such as true positive (TP), false positive (FP), and false negative (FN):(6)$$\mathrm{Recall}=\frac{T\;P\;}{T\;P\;+\;F\;N\;}$$
(7)$$\mathrm{Precision}\;=\frac{T\;P\;}{T\;P\;+\;F\;P\;}$$

**Intersection over Union:** The IoU is a standard metric for evaluating segmentation models. The equation presents the similarity between the predicted pixels (Y′) and the true mask (Y):(8)$$\mathrm{IoU}(Y^{\prime },\;Y)\;=\;\frac{\;\left|Y^{\prime }\;\cap \;Y\right|\;}{\left|Y^{\prime }\right|\;+\;\left|Y\right|\;}\;=\;\frac{T\;P\;}{T\;P\;+\;F\;P\;+\;F\;N\;}$$

**Dice similarity coefficient:** The Dice similarity coefficient is a standard metric for comparing the pixel-wise results between the ground truth and predicted segmentation. The formula of the Dice coefficient is defined as:(9)$$\mathrm{Dice}\;\mathrm{coefficient}(Y^{\prime },\;Y)\;=\;\frac{2\;\times \;\left|Y^{\prime }\;\cap \;Y\right|\;}{\left|Y^{\prime }\right|\;+\;\left|Y\right|\;}\;=\;\frac{2\;\times \;T\;P\;}{2\;\times \;T\;P\;+\;F\;P\;+\;F\;N\;}$$
where Y′ is the predicted set of pixels, and Y signifies the ground truth of the item.

### 4.4. Implementation Details

We trained all the methods in the Keras framework [50] with TensorFlow [51] as a backend. We trained our model with 224 × 224-pixel images. We set the batch size to 10, and we trained the model for 100 epochs. We used the Adam optimizer with reduced learning rate callback; the learning rate starts from 0.01 and is divided by 10 when the patience level exceeds 5. We used an early-stop mechanism on the validation set to avoid overfitting. We chose ReLU as the non-linear activation and binary cross-entropy as the loss function. To convert the predicted pixels to the background or foreground, we used a threshold value of 0.5. All the models were implemented using two NVIDIA GTX 1080 GPUs, each with 8 GB of memory. It took five hours to complete the training of our proposed model.

## 5. Results

We performed comprehensive experiments to assess the effectiveness of our proposed A-DenseUNet architecture. Four state-of-the-art deep learning models, U-Net [10], wide U-Net [11], ResUNet [40], UNet++ [11], PraNet and ResUNet++ were selected for comparative analysis of the proposed method.

To show the effectiveness of each added block we removed one by one each of them. Figure 7 presents the train and validation IoU score of A-DenseUNet and A-DeseUNet without particular blocks, it shows that each block plays a crucial role to achieve better segmentation results. A comparison between the model trained with and without attention blocks shows that the model with an attention mechanism demonstrated strong attention ability by emphasizing the discriminative region of interest rather than concentrating on specular reflection and the normal area. Figure 8 depicts the qualitative difference between them.

Kvasir-SEG dataset results: We improved our proposed A-DenseUNet architecture with various sets of hyperparameters. During the model training, we manually tuned the hyperparameters with various hyperparameter sets and evaluated the results. Table 2 presents the results of A-DenseUNet, ResUNet [40], UNet++ [11], wide U-Net, original U-Net [10], PraNet [52] and ResUNet++ [6] on the Kvasir-SEG [48] dataset. The data indicate that the proposed architecture outperformed all current methods.

CVC-612 dataset: After achieving good results on the Kvasir-SEG dataset, we tested our method on the CVC-612 dataset. Table 3 presents the performance of all the models on the CVC-612 dataset. The proposed method achieved the largest Dice coefficient, IoU, and recall. U-Net obtained the highest precision score, but its other important metric scores for segmentation were not competitive.

Figure 9 and Figure 10 present the qualitative results for all deep learning methods. Table 2 and Table 3 and the qualitative results show the dominance of A-DenseUNet over the baseline methods such as UNet++ [11], ResUNet [40], wide U-Net, original U-Net [10], PraNet [52] and ResUNet++ [6]. On the Kvasir-SEG dataset, the proposed architecture achieved mean improvements of 10.64%, 12.21%, 14.4%, 20.23%, 1.2%, and 9.2% as measured by the Dice coefficient, and 14.12%, 32.21%, 15.03%, 29.86%, 2.15% and 6.81% as measured by the IoU score. The large margin of difference between the proposed architecture and the existing methods could be interpreted as indicating that the combination of dilated convolution, attention mechanism, and multiscale features plays a crucial role in optimizing segmentation efficiency. The proposed model encodes multiscale semantic information at every stage, which allows the conservation of more fine-grained feature maps at the decoder block, unlike U-Net and ResUNet, which use the same-scale feature map concatenation. Furthermore, the attention mechanism enhances the network by focusing on important information that boosts the segmentation efficiency.

## 6. Discussion

The proposed A-DenseUNet model achieved adequate results on both the CVC-612 and Kvasir-SEG datasets. From the qualitative results, it is obvious that the proposed model’s segmentation mask performed better than other methods to capture the shape of information on the Kvasir-SEG dataset. The results show that the predicted segmentation mask in A-DenseUNet is closer to the ground truth mask than that in other state-of-the-art architectures. However, segmentation masks from UNet++ and wide U-Net are also competitive.

During the training process, we used various loss functions to improve our results, such as Jaccard loss, Dice loss, mean square loss, and binary cross-entropy loss. According to our experiments, the method achieved a better Dice coefficient value with all loss functions, whereas the IoU score was higher with a binary cross-entropy loss function. We chose the binary cross-entropy loss function based on our analytical assessment. In addition, we found that the number of kernels, batch size, optimizer, loss function, and depth of the model may affect the result.

We speculate that the efficiency of the model could be further improved by enlarging the dataset size, using various augmentations, and adding certain post-processing steps. DenseNets model allows us to design a very deep neural network architecture to achieve significant performance, also the attention mechanism helps to reduce undesirable features like an artifact, specular reflection, and noises. Due to the dilated convolution, we can accelerate run-time with fewer parameters and achieved better segmentation results. We conclude that the A-DenseUNet should not only limited to biomedical image segmentation but could also be expanded to natural image segmentation and other pixel-wise classification tasks. We used all our experience and knowledge to optimize the model, but there might be further optimizations, which could affect the performance of the method.

## 7. Conclusions

In this paper, we have presented an end-to-end biomedical image segmentation architecture, A-DenseUNet, to achieve more accurate segmentation results. This method aggregates multiscale semantic information to generate a global feature and encode such features to the decoder side alongside skip-connection features. This enables the model to learn multi-scale semantic features at each level instead of learning same-scale feature maps. While the attention mechanisms have been widely accepted and used in other computer vision fields, a similar adoption is needed in the medical domain. Hence, we propose a complex model with an attention mechanism that takes inputs from the global module, encoding, and up-sampling features and filters noisy and ambiguous regions effectively and able to capture long-range dependencies in the image. To evaluate the effectiveness of our approach we conduct experiments on the CVC-612 and Kvasir-SEG datasets. The results demonstrate that the proposed method outperforms the state-of-the-art UNet++, ResUNet, and U-Net architectures in predicting accurate segmentation masks. Our model achieved the best Dice coefficient and IoU score among the models. Future work could be included to evaluate our model on other medical and natural image segmentation domains. Also, we want to implement our model in real-time surgical robots as Aleks, et al. [21] and Alessandro et al. [14] suggest in their work.

## Figures and Tables

**Figure 1 sensors-21-01441-f001:**
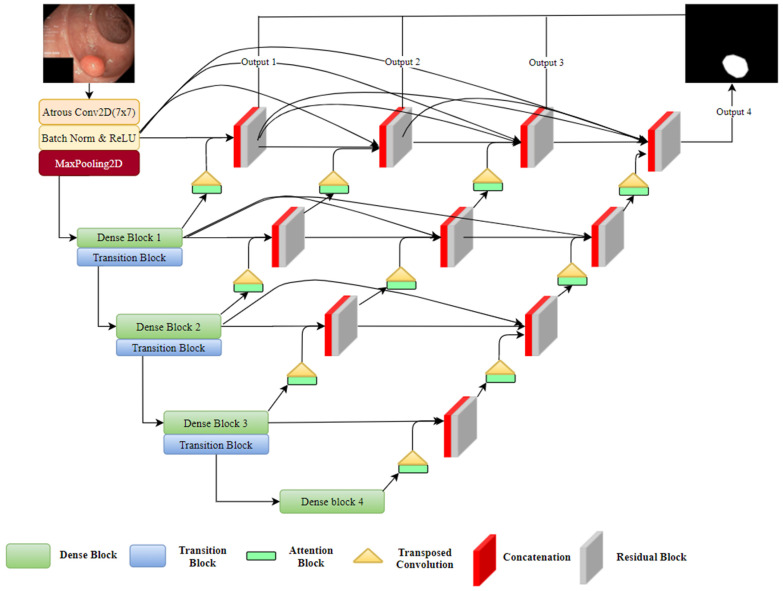
Block diagram of the proposed A-DenseUNet architecture: DenseNet is used as an encoder, Transposed convolution is performed for up-sampling between levels.

**Figure 2 sensors-21-01441-f002:**
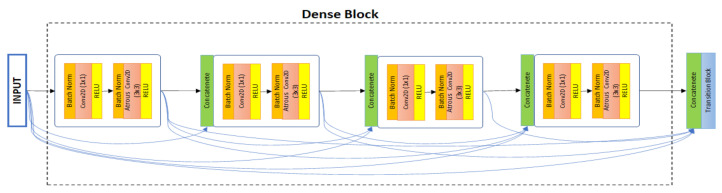
Five-layer dense block with grows rate n = 4. Each layer takes all previous information and includes batch normalization, atrous convolution, and ReLU activation.

**Figure 3 sensors-21-01441-f003:**
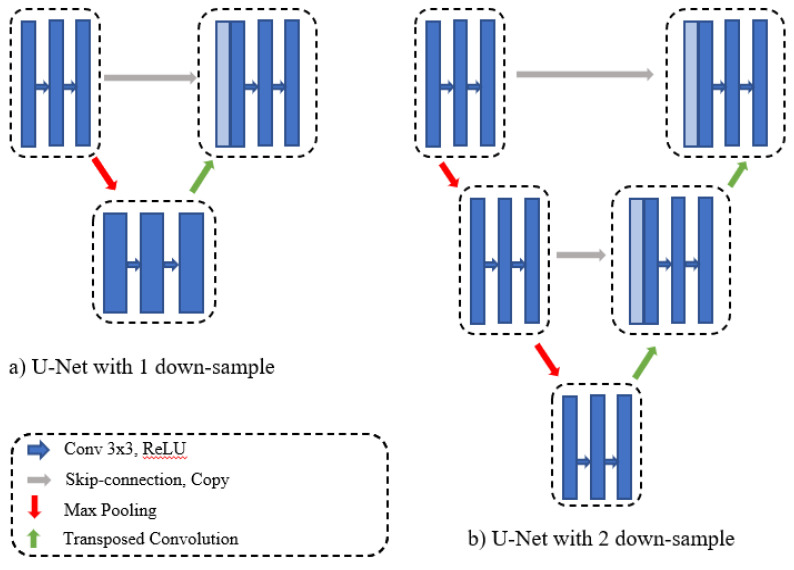
Multi-depth U-Net models.

**Figure 4 sensors-21-01441-f004:**
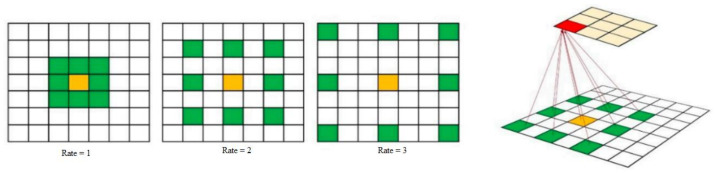
Dilated convolutions with different dilation rates. A dilation rate of one is normal convolution.

**Figure 5 sensors-21-01441-f005:**
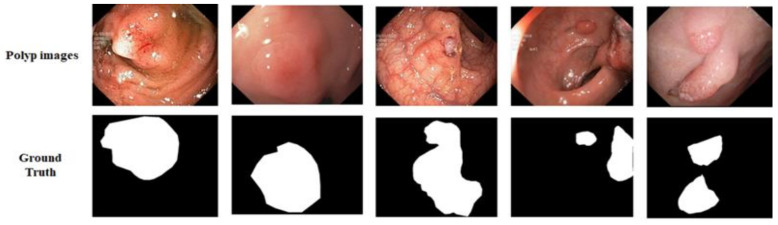
Example of data from Kvasir-SEG dataset. The first row shows original images and the second row presents their respective ground truth.

**Figure 6 sensors-21-01441-f006:**
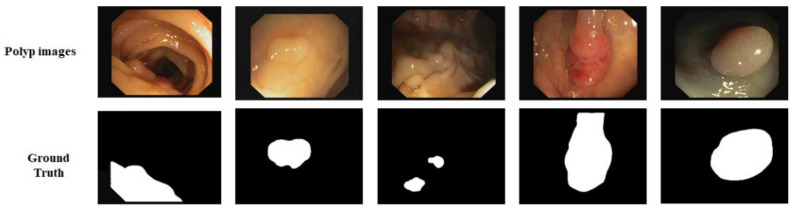
Images and ground truth masks from the CVC-612 dataset.

**Figure 7 sensors-21-01441-f007:**
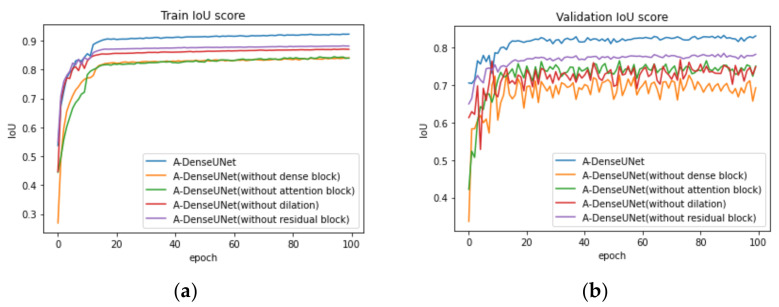
(**a**) Train and (**b**) validation IoU score after one by one removing added blocks.

**Figure 8 sensors-21-01441-f008:**
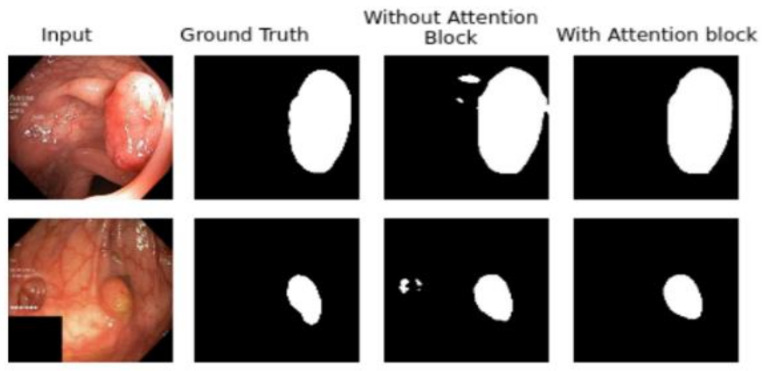
Effect of attention block in the network. By adding this, we were able to suppress the irrelevant regions.

**Figure 9 sensors-21-01441-f009:**
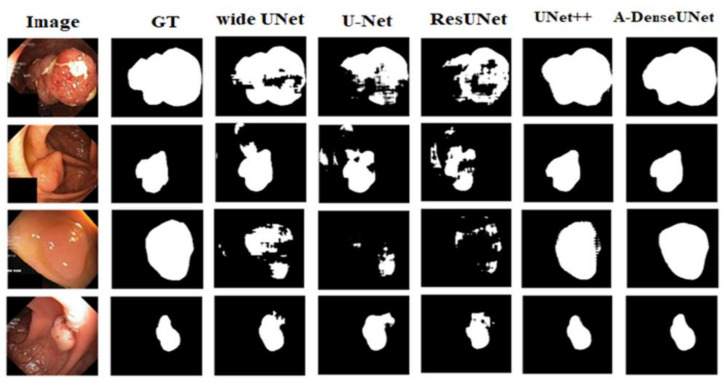
Qualitative segmentation results of various models on the Kvasir-SEG dataset. Experimental results show that A-DenseUNet produces better segmentation masks than other state-of-the-art networks.

**Figure 10 sensors-21-01441-f010:**
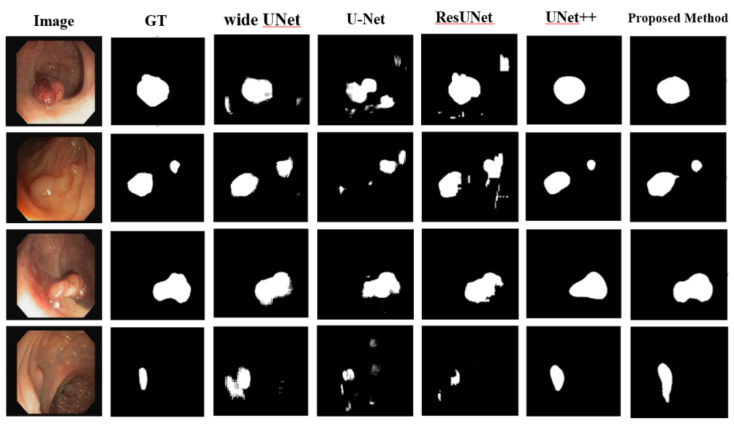
Qualitative segmentation results of various models on the CVC-612 dataset.

**Table 1 sensors-21-01441-t001:** Densely connected encoder block of the proposed A-DenseUNet architecture. Note that “1 × 1, 192 conv” corresponds to 1 × 1 kernel size convolution with 192 features and a sequence of BN-Conv-ReLU layers. “[] × n” indicates n iterations of the dense block.

	Feature Size	Encoder DenseNet-164 (k = 48)
input	224 × 224 × 3	-
convolution 1	112 × 112	7 × 7, 96, stride 2
pooling	56 × 56	3 × 3 max pool, stride 2
dense block 1	56 × 56	$\left[\begin{array}{c} 1\;\times \;1,\;192\;\;\mathrm{conv}\\ 3\;\times \;3,\;48\;\;\mathrm{conv} \end{array}\right]$ × 6
transition layer 1	56 × 56	1 × 1 conv
	28 × 28	2 × 2 average pool, stride 2
dense block 2	28 × 28	$\left[\begin{array}{c} 1\;\times \;1,\;192\;\;\mathrm{conv}\\ 3\;\times \;3,\;48\;\;\mathrm{conv} \end{array}\right]$ × 12
transition layer 2	28 × 28	1 × 1 conv
	14 × 14	2 × 2 average pool, stride 2
dense block 3	14 × 14	$\left[\begin{array}{c} 1\;\times \;1,\;192\;\;\mathrm{conv}\\ 3\;\times \;3,\;48\;\;\mathrm{conv} \end{array}\right]$ × 36
transition layer 3	14 × 14	1 × 1 conv
	7 × 7	2 × 2 average pool, stride 2
dense block 4	7 × 7	$\left[\begin{array}{c} 1\;\times \;1,\;192\;\;\mathrm{conv}\\ 3\;\times \;3,\;48\;\;\mathrm{conv} \end{array}\right]$ × 24

**Table 2 sensors-21-01441-t002:** Kvasir-SEG evaluation results of all methods.

Method	Params	Dice	IoU	Recall	Precision
**A-DenseUNet**	**11.0 M**	**0.9085**	**0.8615**	**0.9448**	**0.9766**
UNet++ [11]	9.0 M	0.8021	0.7215	0.7914	0.9321
ResUNet [40]	8.5 M	0.7864	0.5421	0.7861	0.8912
Wide U-Net [11]	9.1 M	0.7645	0.7112	0.7684	0.9231
U-Net [10]	7.8 M	0.7062	0.5628	0.7768	0.9022
PraNet [52]	-	0.898	0.84	-	-
ResUNet++ [6]	-	0.8133	0.7927	0.7064	0.8774

**Table 3 sensors-21-01441-t003:** CVC-612 evaluation results of all methods.

Method	Params	Dice	IoU	Recall	Precision
**A-DenseUNet**	**11.0 M**	**0.8912**	**0.8553**	**0.9448**	0.9266
UNet++ [11]	9.0 M	0.7815	0.7241	0.8064	0.9076
ResUNet [40]	8.5 M	0.7397	0.5597	0.7643	0.8627
Wide U-Net [11]	9.1 M	0.7754	0.7078	0.7831	0.9113
U-Net [10]	7.8 M	0.6943	0.5798	0.7648	**0.9418**
PraNet [52]	-	0.89	0.849	-	-
ResUNet++ [6]	-	0.7955	0.7962	0.7022	0.8785

## Data Availability

Not applicable.

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
