# Peer review of "A-DenseUNet: Adaptive Densely Connected UNet for Polyp Segmentation in Colonoscopy Images with Atrous Convolution"

_sensors, 2021, doi:10.3390/s21041441_

Round 1
Reviewer 1 Report
Comments on the paper “
A-DenseUNet: Adaptive Densely Connected UNet for Polyp Segmentation in Colonoscopy Images with Atrous Convolution“
Content comments
- This paper presents an interesting proposal, but some points must be improved before the paper is approved.
- The novelty aspects of the present research must be presented explicitly (show your improvements in Fig.1). More details should be furnished.
- The proposed approach is presented in a very abstract way. Therefore, the description of the proposed approach must be changed in a more detailed way stressing on the novelty aspects. More details should be furnished.
- The eq.(1,2, 4,5) must have the same notation as the description of the approach (indices are omitted).
- In the case study section the results in table 3 do not correlate to results presented in Fig.10. More details should be furnished.
- The conclusions must be more closely directed to the novelty aspects of the paper.
Author Response
Dear reviewer thank you to review our article.
We replyed to all the given points in the attached document file.
Please see the attachment.

Reviewer 2 Report
The authors have analysed CVC-612 and Kvasir-SEG dataset for segmentation of polyps in colonoscopy images using various model and it seems A-DenseUNet model interprets better polyp image. This study will have broader impact in clinical use. Statistical analysis is missing in this study. How many images are analysed in each model and whether in all the cases the new model has significantly better resolution. The images are from one patient or multiple random?
Author Response
Dear reviewer thank you to review our article.
We replied to all the given points in the attached document file.
Please see the attachment.

Reviewer 3 Report
The manuscript "A-DenseUNet: Adaptive Densely Connected UNet for Polyp Segmentation in Colonoscopy Images with Atrous Convolution" presents a CNN for polyp segmentation in colonoscopy images. The manuscript addresses a relevant problem and is easy to follow.
I have some major concerns that should be addressed prior considering the work for publication:
1) The authors present a very complex architecture, with attention mechanism, atrous convolution, residual and dense blocks. Are all these architectural elements needed? An ablation study is needed to show the segmentation results when removing one by one all the architectural elements.
2) The authors use publicly available datasets. A table should be added to compare the achieved performance with that of methods in the literature using the same dataset(s).
3) The discussion should be improved highlighting the benefit of each of the architectural elements included in the network.
4) The part on future work should be improved. For example, have the authors considered processing the temporal information encoded in colonoscopy videos? Research in the endoscopic field already exists (even if for different aim), e.g. Attanasio, Aleks, et al. "A Comparative Study of Spatio-Temporal U-Nets for Tissue Segmentation in Surgical Robotics." IEEE Transactions on Medical Robotics and Bionics (2021). Similarly, adversarial training could help tackling some of the challenges of the endoscopic dataset, see Casella, Alessandro, et al. "Inter-foetus membrane segmentation for TTTS using adversarial networks." Annals of Biomedical Engineering 48.2 (2020): 848-859.
Author Response

(The authors gave the same response as above.)

Round 2
Reviewer 1 Report
The manuscript can be published.
Author Response
We are very grateful to the reviewer to give permission to publish. We appreciate it and it motivates us with broad confidence to form the right direction and investigation more research topics in the future.
Reviewer 3 Report
Thank you for revising the manuscript.
Unfortunately, some of my concerns were not fully addressed.
Point 2) It is not clear if other researchers in the literature tested their methods on the same dataset used in this work. If so, the performance of these methods should be compared to that achieved in this manuscript.
Point 4) The authors did not add any methodological references in the future work part (I have suggested a couple of them). It is difficult for the reader to understand where to search in the literature to advance the results presented in this manuscript.
Round 3
Reviewer 3 Report
Thank you for revising the manuscript.